# Mapping the Urban Population in Residential Neighborhoods by Integrating Remote Sensing and Crowdsourcing Data

**Chuanbao Jing [1,2], Weiqi Zhou [1,2,3,*] , Yuguo Qian [1] and Jingli Yan [4]**

[1]  State Key Laboratory of Urban and Regional Ecology, Research Center for Eco-Environmental Sciences, Chinese Academy of Sciences, Beijing 100085, China; cbjing_st@rcees.ac.cn (C.J.); ygqian@rcees.ac.cn (Y.Q.)
[2]  College of Resources and Environment, University of Chinese Academy of Sciences, Beijing 100049, China
[3]  Beijing Urban Ecosystem Research Station, Research Center for Eco-Environmental Sciences, Chinese Academy of Sciences, Beijing 100085, China
[4]  Division of Forest, Nature and Landscape, KU Leuven, 3001 Heverlee, Belgium; jingli.yan@kuleuven.be
*  Correspondence: wzhou@rcees.ac.cn; Tel.: +86-10-6284-9268; Fax: +86-10-6291-5372

**Abstract:** Where urban dwellers live at a fine scale is essential for the planning of services and response to city emergencies. Currently, most existing population mapping approaches considered census data as observational data for specifying models. However, census data usually have low spatial resolution and low frequency. Here, we presented a framework for mapping populations in residential neighborhoods with 30 m spatial resolution with little dependency upon census data. The framework integrated remote sensing and crowdsourcing data. The observational populations and number of households at residential neighborhood scale were obtained from real-time crowdsourcing data instead of census data. We tested our framework in Beijing. We found that (1) the number of households from a real estate trade platform could be a good proxy for accurate observational population. (2) The accuracy of the mapping population in residential neighborhoods was reasonable. The mean absolute percentage error was 47.26% and the $R^2$ was 0.78. (3) Our framework shows great potential in mapping the population in real time. Our findings expand the knowledge in estimating urban population. In addition, the proposed framework and approach provide an effective means to quantify population distribution data for cities, which is particularly important for many of the cities worldwide lacking census data at the residential neighborhood scale.

**Keywords:** urban population estimation; remote sensing; fine-scale; census; dasymetric mapping; nighttime light

## 1. Introduction

Population distribution data are critical for planning, governance, and research in urban areas [1–3]. The spatiotemporal resolution of population distribution data affects decision-making and planning [4], the ability to address city emergencies (e.g., the Corona Virus Disease 2019, earthquakes, and tsunamis) [5,6], the credibility of some studies (e.g., social equity [7], disaster assessments [8], and site selection [9]), etc. The most commonly used population data in the world are official census data, which are always obtained at a low spatial resolution and low frequency (even though the data are unavailable in some poor or politically unstable countries) [2]. However, the population distribution in an urban area exhibits strong heterogeneity. This coarse spatiotemporal resolution of census cannot match the strong heterogeneity of the urban population [10,11]. Consequently, a lack of precise population data has become an obstacle for governors, planners, and researchers in heterogeneous urban areas.

The increasing power of geographic information systems (GISs), massive remote sensing products, and multiple geospatial big data have improved population estimations, especially at the regional and global scale [5]. Such advances have greatly improved the spatiotemporal resolution and the standards of population maps. Additionally, improved classification technology (e.g., land cover/land use) and many emerging ancillary data have also greatly contributed the improvement of population mapping. Many population datasets have been developed and are commonly used (Table 1) [5]. Such datasets include the Gridded Population of the World (Gpw4.11), Worldpop, and the LandScan Global Population Database (LandScan Global) [5,12,13]. In addition, some studies focus on populations in small areas, with spatial resolution less than 100 m (Table 1) [14–16]. These cases lay a good foundation for developing and improving precise population data in urban areas.

**Table 1.** Detailed characteristics of population data at different scales.

|  | Method | Spatial Resolution | References |
|---|---|---|---|
| Global-scale | Areal interpolation (Dasymetric mapping) | 100–1000 m | Balk et al. 2006; Bhaduri et al. 2007; Leyk et al. 2019 |
| National/Regional-scale | Areal interpolation (Dasymetric mapping) | More than 100 m | Li and Zhou 2018; Azar et al. 2013; Deville et al. 2014 |
| Local-scale | Statistical modeling approach | Less than 100 m/ block level | Dong et al. 2010; Silvan-Cardenas et al. 2010; Weber et al. 2018; Wang et al. 2019 |

Note: The blocks are commonly generated by enclosed transportation networks, which always contain several residential neighborhoods.

Among these cases, the population mapping approaches can be summarized into two categories: statistical model and areal interpolation. Statistical modeling approach is a historical and census-independent population mapping approach. This approach is designed to address the shortcomings of a census, including its high cost, low frequency, low spatial resolution, and labor intensity [2,17]. Statistical model estimates population directly via the relationship between the population and population-related factors using statistical models, such as Ordinary Least Squares (OLS) regression, geographically weighted regression, random forest (RF), and neural networks, in small areas [14–16,18]. Samples in small areas are obtained by field surveys or a partial census [2]. Although this approach directly estimates the population of small areas or high spatial-resolution grids, the total population in administrative units can also be estimated by aggregating these high-resolution predictions [2]. However, there are some differences between the aggregated population data and census results. For example, the aggregated population in a country using the statistical modeling approach is always slightly different from the census results in the country.

Dasymetric mapping is a typical areal interpolation approach [17,19,20], and some enhanced forms have been developed, including those employing empirical sampling for each class [21], limiting variable estimation [3], and regressions [22]. Assuming that people do not die and are not born, these approaches redistribute the population from the source units (high level population units) to the target units (a low level population unit) using population-related ancillary information [19,22–24]. Ancillary information, such as land use/land cover data, can provide a good proxy for where people live and how many people live there [18,22]. The spatial heterogeneity of populations in a source unit is expressed by this ancillary information with a weight coefficient ($W_i$) in the disaggregation process, and therefore the population in the source units is allocated to the target units according to $W_i$. The general formula is as follows,

$$P_i = \frac{P \times W_i}{\sum_{i=1}^{n} W_i} \quad (1)$$

where $P$ is the total population in a source unit (the population to be allocated), $P_i$ is the population of the ith target unit or pixel in the source unit, $W_i$ is the population-based weight of the ith target unit or pixel, and n is the total number of target units or pixels. How to obtain $W_i$ from ancillary information is the most controversial and crucial factor for this approach [20]. Earlier, $W_i$ was generated by land use/land cover data [25]. Uninhabited and inhabited areas (a binary classification) in a large region were first marked according to their land cover. Then, the population in a large region was redistributed to inhabited areas and aggregated at the subdistrict level. Later, statistical model was introduced to dasymetric mapping to quantify the relationship between ancillary information and the population [26]. Since then, defining $W_i$ via the initial estimated value from statistical model has become prevalent [20,22]. However, census data are the only effective and available population data for source units, and some flaws of the population census data limit the value of dasymetric mapping. For example, the precise population or density is controlled by the difference in scale between the source units and target units, and the temporal resolution of a population in a target unit is controlled by that of the census, which is always low (e.g., 10 years in China at the subdistrict level).

With these two approaches, many ancillary information sources or factors related to population have been discussed and used to estimate population. Satellite imaging products, given their high resolution and short acquisition cycle, have been the most common ancillary datasets used to map the populations, e.g., land cover/land use data and nighttime light (NTL) imagery, over the past few decades [18,22,27,28]. Land cover/land use data represent human modification of the earth's surface. Different land cover/land use always have different population densities [22]. NTL can serve as an indicator of human activities [27]. High human activity is more likely correlated with a high NTL value. However, these satellite image products generally have a coarse resolution, which cannot reflect the strong heterogeneity in a city; these products have limited capabilities to reveal the socioeconomic features related to the population distribution [29–31]. In recent years, some geographical data have showed great potential for mapping populations at a fine sale. Examples include building features (e.g., the number of buildings, floor area, and building area), road density, location, phone calls, and location-based service data, such as points of interests (POIs), traffic card information, Tencent user data, and mobile phone data [1,18,23,32–35].

The population distribution in urban areas exhibits strong heterogeneity [11,18]. However, only a few of studies have been estimated at a fine local scale (less than 100 m) in urban areas. In addition, regression is the most commonly adopted method in both the statistical modeling approach and dasymetric mapping, and observational populations are the dependent variable in a regression. For accurately estimating urban population, an accurate observational population are indispensable. Recent studies at fine local scale had showed some means of obtaining population samples data, but most of them were hard to collect for general researchers. These studies always obtained population samples from labor-intensive surveys and expensive or biased Tencent user data [16,18]. As a compromise, population samples at a large scale were used to construct a model and estimate results at a finer scale in some studies [1,11,20,29]. Note that the estimated populations from these statistical downscaling approaches may be biased. This problem arises because the rules may change at different scales, which is a widely accepted phenomenon in ecology and geology [36–39]. Therefore, a lack of accurate population samples for general researchers has become the primary limitation of estimating accurately population in fine local scale. Moreover, some population-related factors were used to map the urban population, such as land use/land cover, NTL imagery, and building structures [1,18,28]. However, at a fine local scale, the relationships between these factors and population require further confirmation, and the relative importance of the factors remains unclear.

By addressing these gaps in population mapping for urban areas, we present a new approach that integrates remote sensing and crowdsourcing data to map urban population at residential neighborhood level (Figure 1). Populations in a residential neighborhood in this paper are the people who live in the residential neighborhood, which is similar to household survey data (census data). A proxy for accurate observational population (the number of households in a residential neighborhood),

crowdsourcing data from a real estate trade platform, was first used as the sample for regression. Ten population-related factors from five aspects—the building features, activity, environment, location, and traffic—were discussed and employed. In this study, we (1) revealed the relationships between these population-related factors and population and the relative importance of the population-related factors at the residential neighborhood level, and (2) estimated populations in residential neighborhoods in the Beijing fifth-ring area using statistical modeling approach and generated a gridded prediction of population density at 30 m spatial resolution. A flowchart of this approach is shown in Figure 1. The approach can generate fine-scale population information and closely match the strong heterogeneity of an urban population.

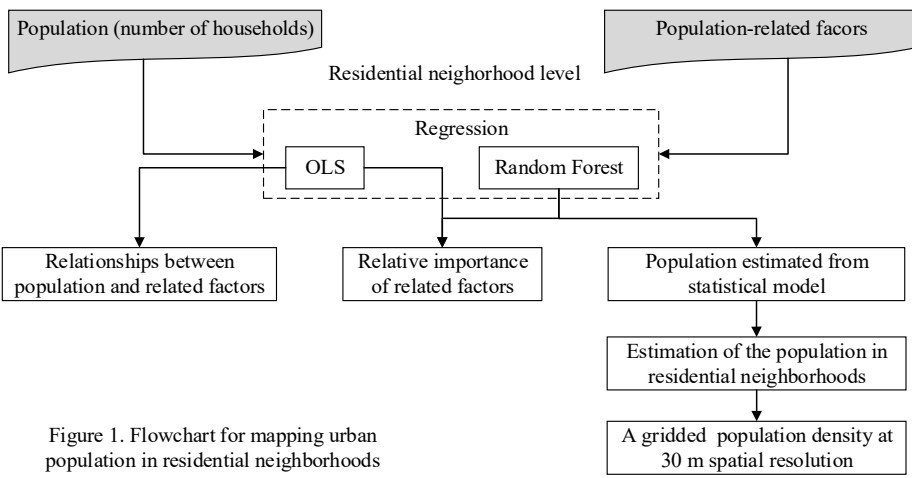

Figure 1. Flowchart for mapping urban population in residential neighborhoods

**Figure 1.** Flowchart for mapping urban population in residential neighborhoods.

## 2. Study Area and Data

### 2.1. Study Area

We focused our study on areas within the fifth-ring road of Beijing, which is the core urban area of the city. Beijing is a megacity and the capital of China. The total area of Beijing is 16,410 km$^2$, and it contains 16 districts (equivalent to level 3 of the Global Administrative Unit Layer defined by the Food and Agriculture Organization) and 325 subdistricts (equivalent to level 4 of the Global Administrative Unit Layer). The population data from the government have relatively coarse temporal and spatial resolutions. Two types of population data were generated. The first type of data is from a population census every 10 years at the subdistrict level, and the second type is from a population sampling survey conducted every year at the country level. From the population sampling survey in 2018, the permanent population was estimated to be 21.54 million and the population density was 1313 people per square kilometer (Beijing statistical yearbook, 2019). Beijing has a typical concentric circle structure because of its long history. The population density follows a roughly decreasing trend from the center to the suburbs. The fifth-ring area is the core area of Beijing and is the most densely populated area containing nearly half of the population in 4% of the area (Beijing Municipal Bureau of Statistics, 2016). Although the fifth-ring area is a deeply rooted concept that has been embedded in residents' lives and one of the most popular study areas, the population survey data from 2015 are the only record of this area because of the mismatched boundary between the fifth-ring area and the administrative unit. Therefore, we selected the fifth-ring area in Beijing as our study area (Figure 2).

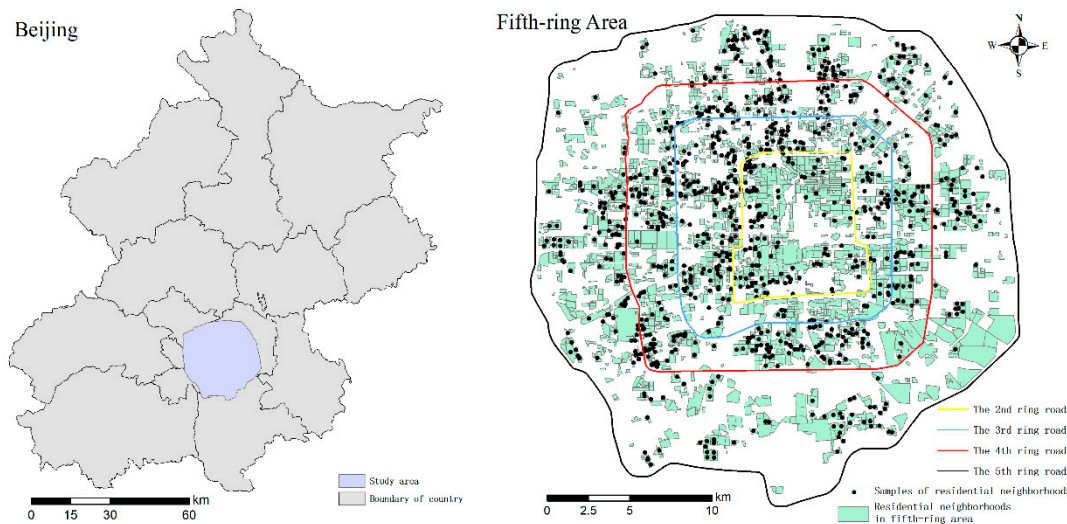

**Figure 2.** Study area within the Fifth-ring road in Beijing, the urban core areas of Beijing.

## 2.2. Data and Processing

Residential neighborhoods, which were used as our mapping unit, were visually interpreted at a 1 m spatial resolution and were generated using very high-resolution imagery (GeoEye) in 2009 and the POIs of residential neighborhoods from the BaiduTM map data (map.baidu.com) in 2010. A POI provided the location and name of the residential neighborhood located at a point. The GeoEye data were further used to identify the boundaries of residential neighborhoods via visual interpretation. Finally, 3164 total residential neighborhoods (Figure 2) were mapped, and their areas were calculated.

Multiple ancillary or "covariate" data were employed to quantify the relationship between population and population-related factors and then to estimate the population. In practice, integrating population-related factors from different facets is likely to be able to generate a more accurate estimation of population [2]. Here, we considered population-related factors and collected ancillary data from five aspects: building morphology, human activity, environment, location, and traffic. These five aspects were selected based on their potential impacts on population distribution/density, as reported from previous studies, as well as data availability. For example, previous studies have shown that building morphology generally performs better than land cover for estimating population, as building morphology data can offer more direct residence information about dwellers [15,31]. Environmental factors such as vegetation cover may also affect population distribution due to people's preference on green space [29,40,41]. Gaughan et al. (2016) showed that location is related to population density, indicated by a decay distance of population density from urban core area to urban edge [42]. Additionally, traffic is directly related to dwellers' travel, which affects population distribution on some extent [29]. Building morphology was represented by the area of residential neighborhoods and some features in a residential neighborhood, including the average number of building floors, the building area, the number of buildings, the floor area, and the area of impervious surfaces. The building area is defined as the area of the building footprint or the vertical projected area of the building [43]. The floor area is defined as the total floor area of a building. Activity, environment, location, and traffic data are represented by the annual average nighttime lights, the area of vegetation in a residential neighborhood, the distance from the center of Beijing (distance), and the length of roads around a residential neighborhood, respectively.

The areas of vegetation and impervious surfaces in each residential neighborhood were extracted from land cover classification. The classification was generated from the GeoEye imagery at a 1 m spatial resolution. The classification method was adopted from [44]. The annual nighttime light data in 2012 were aggregated to residential neighborhoods. The original nighttime light data were from the Visible Infrared Imaging Radiometer Suite (VIIRS) day/night band (DNB), which is composed

monthly at a spatial resolution of 450 m. Then, the annual average nighttime light data were calculated using the Google Earth engine and resampled to 30 m [18]. The average number of building floors, the building area, the number of buildings, and the floor area were calculated using building footprints from Amap (map.amap.com), and the boundaries of residential neighborhoods were mapped based on overlay analysis. The road network was collected from Open Street maps (http://openstreetmap.org/) in September 2010. The lengths of the roads in various buffers around residential neighborhoods (0.5 km, 1 km, and 2 km) were calculated. The distance from the center of Beijing (Imperial Palace) was extracted to indicate the location. Except for the distance and length of roads, other variables were measured using their natural logarithm. Log-transformation is commonly used to achieve a normal distribution of population-related factors and therefore obtain more accurate relationships and population [22,43,45].

A total of 943 samples of residential neighborhoods were used to reveal the relationships, the relative importance level of these factors and the population and construct the RF model (Figure 2). When a sample population was absent, proxies for the population were always adopted, such as the number of dwelling units [1], the number of Tencent users [18,32], and mobile phone data [23]. In this paper, we chose the number of households as a proxy for the population in a residential neighborhood. The number of households in a residential neighborhood and their locations were obtained from Lianjia[TM] (one of the most popular real estate trade platforms in China, https://bj.lianjia.com/). Then, the number of households was linked to population-related factors based on location.

In addition, we also used census data at the subdistrict level from the 6th National Population Census of the People's Republic of China 2010 generated by the National Bureau of Statistics of China. The boundaries of the countries and subdistricts were those from 2010.

## 3. Methods

This section is organized as follows (Figure 1). With the number of households in residential neighborhoods as a proxy for the population, we first quantified the relationships between the population and population-related factors using the OLS regression and the relative importance using RF regression with the assistance of the coefficient of determination ($R^2$) value from the OLS results. After correlation analysis, we mapped the urban population at a fine scale using two popular estimation approaches: statistical modeling approach and dasymetric mapping. Finally, the mapping accuracy of the population was assessed.

### 3.1. Relationships between Population and Related Factors and Their Relative Importance

The relationships between population and its related factors from the samples were investigated using OLS regression. Population (dependent variable) is represented by the number of households. The related factors (independent variables) included the areas of vegetation and impervious surfaces, the NTL data, the average number of building floors, the building density, the number of buildings, the building area, and the road length (0.5 km buffer, 1 km buffer, and 2 km buffer). Only one related factor was employed for each OLS model. The $R^2$ was used to measure the robustness of the regression model and the capability of the related factor to explain the population.

The relative importance of the related factors was estimated by RF regression. The percentage-increased mean square error (%IncMSE) and IncNodePurity are common indexes used to measure the importance of variables in the RF model from different aspects [46]. The %IncMSE has been commonly used to measure the increase of the mean squared error after removing a factor from the RF model. A more important factor has a higher %IncMSE. IncNodePurity is the total decrease in node impurities from splitting a variable, which is measured by the residual sum of squares in the regression. In addition, the $R^2$ from the OLS regression estimation can also be used to assess the relative importance of related factors as an auxiliary index.

*3.2. Mapping the Population*

The statistical modeling approach was commonly used to map population in fine local scale, and this approach was employed to map the urban population in this paper. The regression RF model was used to quantify the relationships between the independent variables (the areas of vegetation and impervious surfaces, the NTL data, the average number of building floors, the building area, the number of buildings, the floor area, and the length of roads (1 km buffer)) and the dependent variable (the number of households in residential neighborhoods). RF is a typical nonparametric machine learning approach [46]. The approach creates many individual decision trees and then predicts outcomes based on the average results from all the trees. Due to having minimal assumptions (no need for normality, homogeneous variance, standardized data, and independence between explanatory variables), strong robustness, a powerful approximation ability for a nonlinear function, and few parameters, RF is widely used to map populations [22]. Here, the RF model was used to estimate the number of households in residential neighborhoods. Two-thirds of the samples were employed to train the RF model, and one-third of the samples were used as a validation dataset. Then, a fixed household–population ratio was used to estimate the population using the estimated number of households. The household–population ratio was calculated by the Beijing statistical yearbook 2010 at the country level. Finally, the population in a residential neighborhood was converted to a raster format with 30 m spatial resolution. This spatial resolution is considered reasonable, because it is close to the size of building roofs [18].

For comparison of approach, dasymetric mapping approach also was employed. Generating weights using a statistical model is an important step in this approach. Here, the estimated populations of residential neighborhoods from the RF regression were used as the weights [22]. For each subdistrict (the source unit to be allocated), the population distribution in residential neighborhoods was generated using Equation (2), as follows,

$$P_{ij} = \frac{P \times E_i}{\sum_{i=1}^{n} E_i} \tag{2}$$

where $P_i$ is the population of the ith residential neighborhood, $P$ is the total population of the subdistrict, $E_i$ is the estimated number of households in the ith residential neighborhood from the RF regression, and n is the number of residential neighborhoods in the subdistrict.

*3.3. Accuracy Assessment*

The accuracies at the residential neighborhood level and subdistrict level were both estimated. The 1:1 line, $R^2$, mean square error (RMSE), and mean absolute percentage error (P), which are well-known accuracy assessment metrics of population mapping [1,18,47,48], were selected to comprehensively assess the accuracy at the residential neighborhood scale. The $R^2$ is from a zero-intercept regression. The RMSE and P are calculated using the estimated population from the population mapping model and the actual population. The formulas for these indexes are as follows,

$$\text{RMSE} = \sqrt{\frac{\sum (A_i - E_i)^2}{n}} \tag{3}$$

$$\text{P} = \frac{1}{n} \sum_{i=1}^{n} abs\left(\frac{A_i - E_i}{A_i}\right) \times 100\% \tag{4}$$

where $E_i$ is the estimated population, $A_i$ is the actual population, and n is numbers of samples. At the subdistrict level, only the subdistricts completely located in the fifth-ring area were included because of the limited census by administration units. In addition, the estimated population from dasymetric mapping approach was only used to compare with that from the statistical modeling approach at the residential neighborhood level.

## 4. Results

### 4.1. Relationships between Population and Population-Related Factors

There are various and complex relationships between population and its related factors, including linear relationships, U-shaped relationships, and inverse U-shaped relationships (Figure 3). The area of impervious surfaces, the area of residential neighborhoods, the NTL data, the building area, and the number of buildings exhibited significant positive linear relationships with population. The number of floors exhibited weak positive linear relationships with population. The length of roads exhibited weak negative linear relationships with population. Among three buffers around residential neighborhoods, the length of roads in 1 km buffer exhibited the strongest correlation with population. Therefore, only the length of roads in 1 km buffer was used to represent the traffic factor to estimate population. The floor area and the area of vegetation exhibited significant U-shaped relationships with population. The distance exhibited weak inverse U-shaped relationships with population.

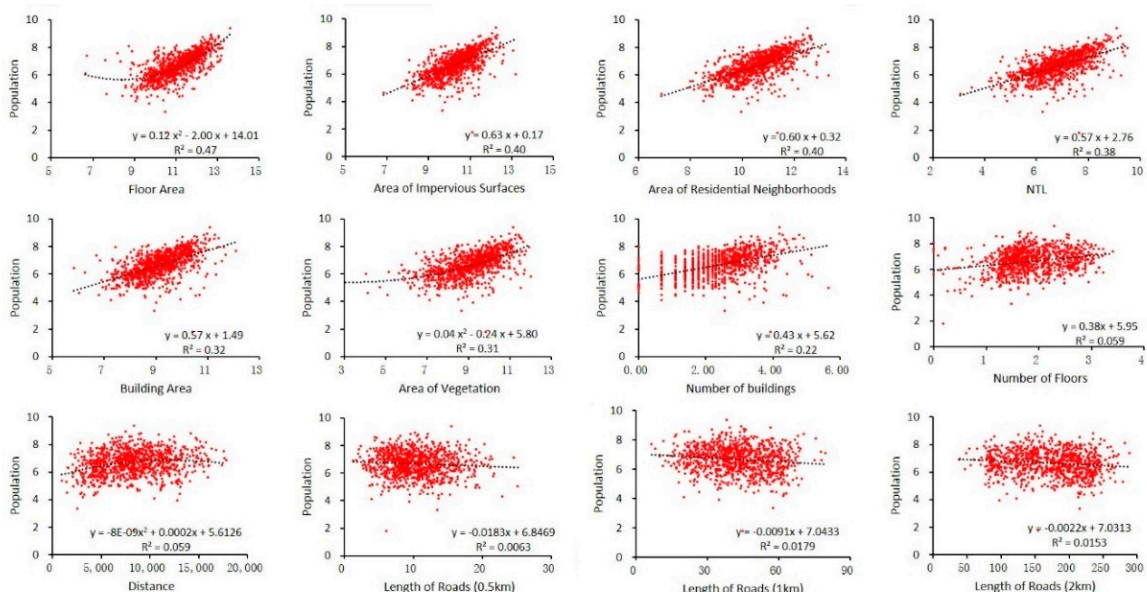

**Figure 3.** Relationships between the related factors and population. Here, population is represented by the number of households in residential neighborhoods, which assumes that the population is proportional to the number of households. All of the variables are measured in natural logarithms so that they followed a normal distribution expect distance and the length of roads, which is normally distributed.

### 4.2. Relative Importance of Population-Related Factors

The relative importance values of the population-related factors in the RF regression are shown in Figure 4. The relative importance values of %IncMSE and IncNodePurity were some different. For example, floor area ranked second for %IncMSE but fifth for IncNodePurity, and area of impervious surfaces ranked fourth for %IncMSE but increased to second for IncNodePurity. However, some similarities could also be found. The indexes of the building features all exhibited the highest importance for %IncMSE and IncNodePurity. The distance and length of roads always had the lowest importance. The $R^2$ values from the OLS regressions are shown in Figure 3. A similar highest importance of building features and similar lowest importances for distance and length of roads were also found for the indexes. Floor area has the highest $R^2$, which was the same for %IncMSE and IncNodePurity.

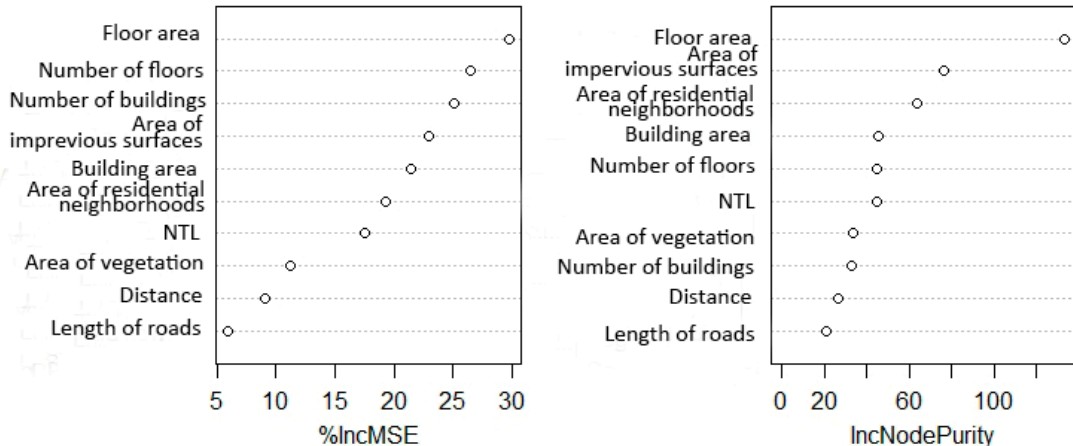

**Figure 4.** Percentage-increased MSE and IncNodePurity indicate the importance of variables in the RF regression model.

### 4.3. Population Accuracy and Methods Comparison

A total of 249 samples (the validation samples in the RF model) were used to assess the accuracy at the residential neighborhood level. The comparison results of dasymetric mapping and statistical modeling approach are shown in Figure 4. Most of the evaluation indexes for population in statistical modeling approach were significantly better than those for dasymetric mapping. Statistical modeling approach had a higher $R^2$ for a zero-intercept regression and a lower RMSE and P than dasymetric mapping (Figure 5). The P of the statistical modeling approach was 15.27% lower than that of the dasymetric mapping approach. However, we found that dasymetric mapping was slightly closer to 1 than the statistical modeling approach.

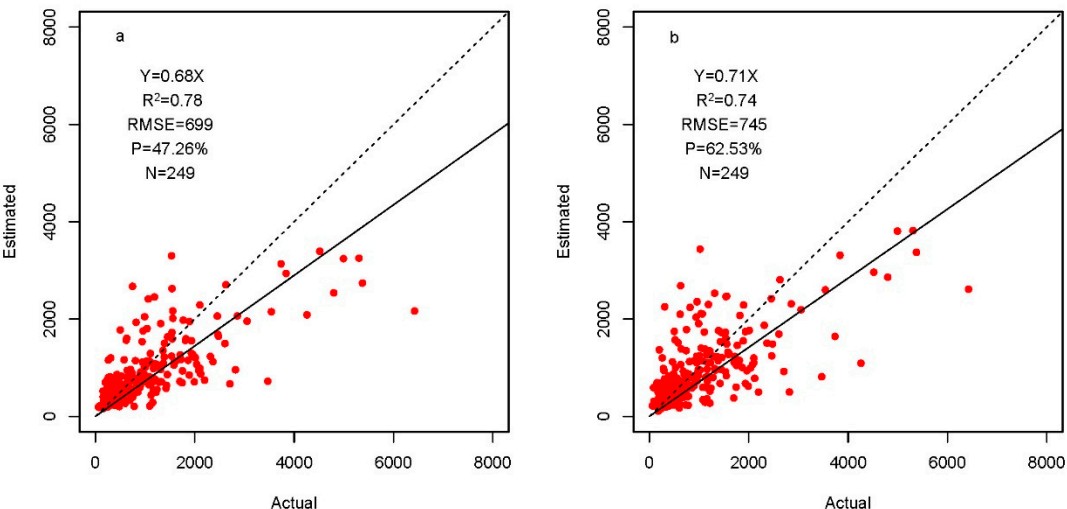

**Figure 5.** Comparison of the two mapping approaches at the residential neighborhood level. (**a**) The accuracy of the statistical modeling approach. (**b**) The accuracy of dasymetric mapping. The black lines are the zero-intercept regression lines.

Supplementary analysis of the accuracy of the statistical modeling approach was carried out at the subdistrict level. A total of 89 subdistricts completely located in the fifth-ring area were employed to assess the accuracy of the estimated population from statistical modeling approach. The accuracies of the number of households and population were similar (Figure 6). The slopes of the zero-intercept regressions were close to 1, 0.84, and 0.87 for the households and population. The $R^2$ values from the zero-intercept regressions were 0.86 and 0.87 for households and population, respectively. P was

approximately 33%. In addition, the RMSE was 12,273 for the number of households and 37,868 for population, respectively.

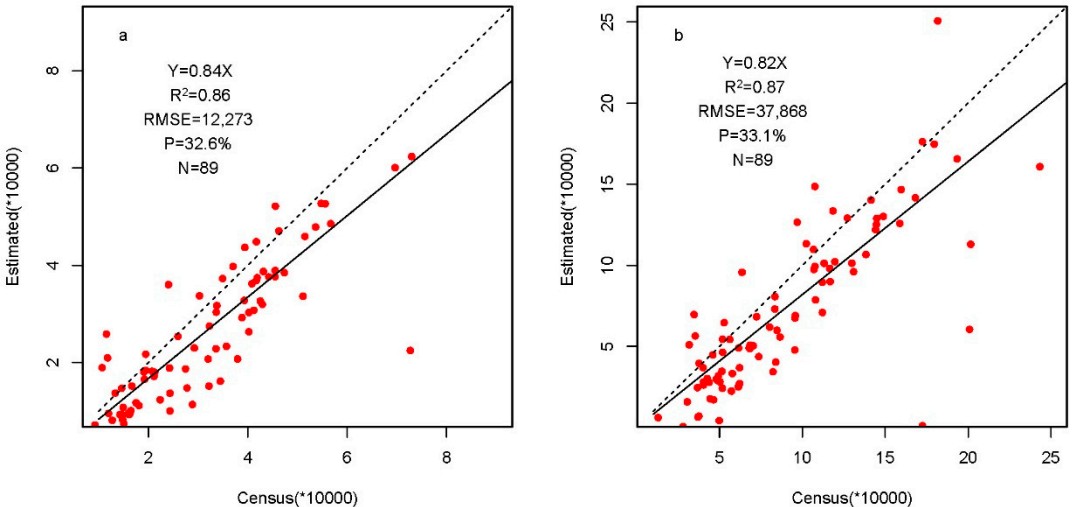

**Figure 6.** Accuracy of statistical modeling approach at the subdistrict level. (**a**) The number of residential households. (**b**) Population. The black lines are the zero-intercept regression lines.

### 4.4. Distribution Features of the Urban Population in the Fifth-Ring of Beijing

The population and population density distribution of the residential neighborhoods in Beijing's fifth-ring area are shown in Figure 7. The populations in various residential neighborhoods exhibited significant differences. The highest population of a residential neighborhood was 17,048 people, and the lowest was only 175 people. The average population of each residential neighborhood was 2337 people. From the center area to the urban fringe, the population of the residential neighborhoods first increased and then decreased. The population was highest around the 3rd ring area.

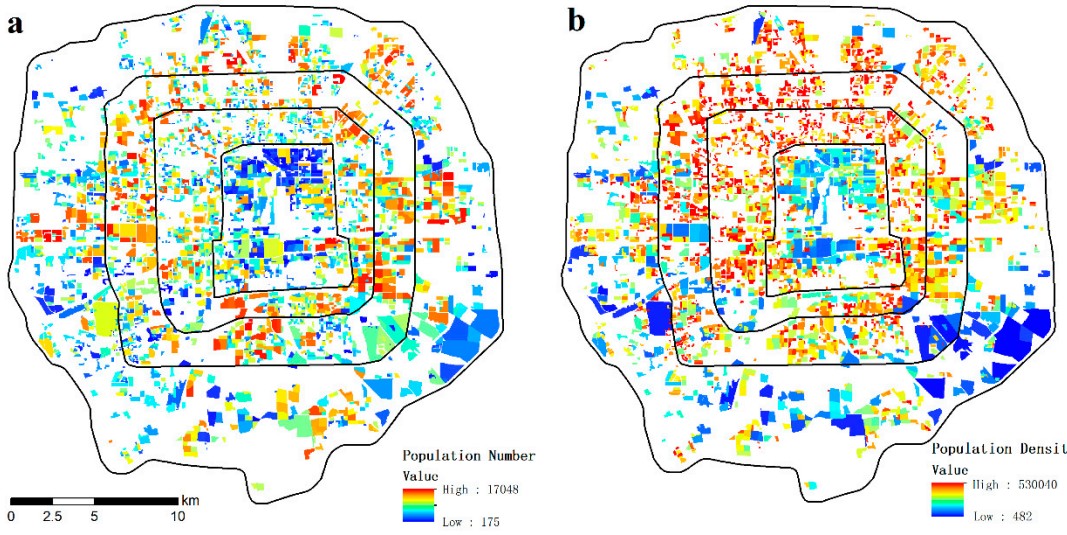

**Figure 7.** Population and population density distribution of the residential neighborhoods in Beijing's fifth-ring area. (**a**) Population in each residential neighborhood and (**b**) population density (people/km$^2$) at the residential neighborhood level.

The populations in the residential neighborhoods of Beijing's fifth-ring area are dense (Figure 7). The average population is 60,770 people/km$^2$ in the residential neighborhoods. Meanwhile, strong spatial heterogeneity can be found in the fifth-ring area. The distribution of the population

density is similar to the distribution of the population. The highest population density occurs around the 3rd ring road, and relatively low population density occurs in 2nd ring area and 5th ring area.

## 5. Discussion

### 5.1. Relationships between Population and Population-Related Factors

The NTL data generally exhibit a significant correlation with population. Moreover, most of the correlations are linear relationships, regardless of whether the dependent variable is population or population density and whether population or population density and NTL are log-transformed [18,20,29,49]. In this paper, we also evidenced a linear relationship between population and NTL after a log-transformation at the residential neighborhood level, but a weaker correlation than that at the large scale (Figure 3) [28]. It should be noted that the seasonal variability of vegetation results in varying lumen output, and may change the trends for some of evaluated relationships [50]. In fact, the same linear relationships between population and other population-related factors can always be found at different levels, but finding a weaker correlation at the residential neighborhood level than at a large level was common [14,15,43]. Exceptions also appeared. For example, Gaughan et al. (2016) assumed that population density decreased as one moved away from an urban center at a large scale. Our result showed that population density first increased and then decreased in Beijing's fifth ring area (Figures 3 and 7). The local policy in Beijing that protects the ancient buildings within the 2nd ring road may have contributed to this result. It should note that we discussed these relationships, irrespective of whether variables are log-transformed. However, the log-transformation is an important step of correlation analysis and OLS regression for non-normal variables. We found the log-transformation was very necessary in exploring the relationships between population and population-related factors, because some variables did not meet normal distribution (Figure 8).

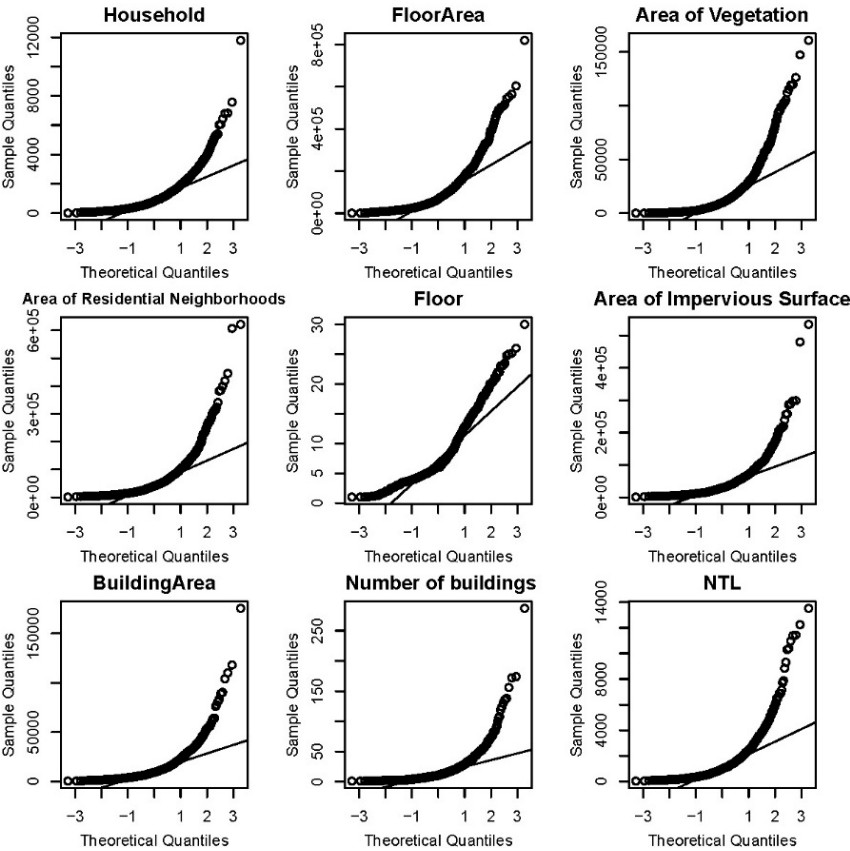

**Figure 8.** Normality test of some variables. Only the non-normal variables were mapped here.

### 5.2. Relative Importance of Population Related Factors

Building features are the most important factors when mapping a population at the residential neighborhood level (Figure 4). This may be because building features can directly reflect the activity of city dwellers related to residence. Residence is the most fundamental demand of dwellers. NTL are the most common data used to map the population at a large scale. We found that NTL exhibited a relatively lower importance at the residential neighborhood level. This may be due to the mismatch of the spatial resolution with residential neighborhoods. The coarse resolution of NTL data may introduce some irrelevant light sources of residential neighborhoods, such as streetlight. The irrelevant lights cannot be removed by resampling. Location, i.e., the distance from the center of the city, was not important compared with the above factors. The long history and high urbanization rate of the study area may be two important reasons for this result. A large number of ancient buildings are located at the urban core (within the 2nd ring area). The number of people in this area is very limited. In addition, the high urbanization rate results in a multicore urban structure. The road length had the lowest importance and a very weak correlation with population for the different buffers. The result may mean that traffic slightly affects the distribution of the urban population.

Scale effects may lead to different rules at different scales. Although similar results have been found at other scales, such as the low importance of traffic and distance at a large scale [29,42], some rules were different. For example, NTL exhibited high importance in estimating the population at a large scale. Gaughan et al. (2016) showed the highest importance of NTL in estimating the population at a large scale (administrative level three, based on the Food and Agricultural Organization framework) [42]. Stevens et al. (2015) showed the second highest importance of NTL at the finest level of the administrative unit (village, Tinh, Kenya) [22]. Moreover, a noninformative contribution of the number of floors was found because many observations had the same number of floors at the block scale [1]. Precipitation was related to the population distribution at a coarse scale but may not be a related factor at a fine scale because of the homogeneity over a small area.

### 5.3. Difference between Dasymetric Mapping and Statistical Modeling Approach

Dasymetric mapping assumes that people do not die and are not born during the redistribution process [24]. This assumption should be matched with a particular administrative unit (source zone). Small source zones have more precise population maps than large source zones. However, even when the smallest source zones (subdistrict) were used, the accuracy of the population was worse than that for statistical modeling approach in residential neighborhoods (Figure 5). The result confirmed Wardrop et al.'s views [2]. The reason for this result may be attributed to the improper redistribution of the residuals in dasymetric mapping. Using the dasymetric mapping performed in this paper as an example, the actual value was equal to the aggregated estimated value plus the aggregated residual for each subdistrict (Equation (5)). Then, the estimated population of the ith residential neighborhood in the jth subdistrict can be expressed as Equation (6). After regression, the residuals should follow a normal distribution and be independent of the RF estimated value in theory. However, in dasymetric mapping, the residuals were redistributed according to the proportion of the estimated value:

$$P_j = E_j + \sigma_j \tag{5}$$

$$P_{ij} = E_j \times \frac{E_{ij}}{\sum_{i=1}^{n} E_{ij}} + \sigma_j \times \frac{E_{ij}}{\sum_{i=1}^{n} E_{ij}} \tag{6}$$

where $P_j$, $E_j$, and $\sigma_j$ are the population, the total estimated population, and the total residuals of the jth subdistrict (the difference between the estimated populations from the RF and the census), respectively. $P_{ij}$ and $E_{ij}$ are the populations of the ith residential neighborhood and the estimated population of the ith residential neighborhood from the RF regression in the jth subdistrict, respectively. n is the number of residential neighborhoods in the jth subdistrict. In addition to its lower accuracy

than statistical modeling approach, dasymetric mapping was also limited by the boundary of the administration units and the temporal resolution of the population data from the government [16]. Obviously, dasymetric mapping does not perform as well at a fine scale as at a large scale (the highest $R^2$ was 0.99 in Bhaduri et al. (2007) [13]). Therefore, it is time to reconsider the value of dasymetric mapping and its assumption at a fine scale.

### 5.4. Real Time Updating of Population Using Statistical Modeling Approach

A national census commonly has a coarse temporal resolution, especially in developing countries [16,28]. The long intervals have limited the application of census-dependent population mapping in a timely manner [51]. Statistical modeling approach is a census-independent approach, which has shown great potential in mapping the population in real time. During the statistical modeling approach, all of the data can be obtained in real-time. We sampled the number of households of residential neighborhoods (dependent variable) from a real estate trade platform. This crowdsourcing data can be updated in real-time. In terms of related factors (independent variables), the number of buildings, the area of impervious surfaces, the area of vegetation, and the building area can be obtained from high frequency satellite imagery [33]. The area of a neighborhood, the floor area and the number of floors can update using remote sensing (e.g., very high-resolution imagery [34]), field investigations or a real estate trade platform. There are generally few new residential neighborhoods in a short time interval. Therefore, it is practical to update residential neighborhoods using a field investigation. POIs and roads can be obtained from open maps (e.g., Baidu$^{TM}$ map and Amap). NTL also has a short revisit period (one month).

### 5.5. Population Data from Different Sources

Census data collected from different scales were most widely used for population mapping worldwide. Obtaining by household survey, population data derived from census data are relatively accurate. However, census data have some limitations [2,22,23]. First, census data usually have relatively low spatial resolution. Although the data are obtained by household survey, only low spatial resolution data (e.g., census blocks in USA and subdistricts in China) can be available for general researchers. Second, census data usually have low frequency. The household survey is commonly carried out every 5 or 10 years. Third, household survey is a time-consuming work. Currently, the population data from remote sensing have become a common supplement of census data. These population data generally have higher spatial resolution than census data [5,23]. In addition, these population data can be obtained in a very short time. For example, using our approach, the population in residential neighborhoods can be updated at any time when needed. However, population data estimated from remote sensing data usually have lower accuracy and higher uncertainty [1,20,28,29]. Therefore, such population data can be complementary to census data, and are particularly useful at fine scale where census data are not available.

### 5.6. Prospects for Future Research

We demonstrated the relationships between population and population-related factors and their relative importance in Beijing's fifth-ring area. In the future, more cases at a fine scale will be needed to verify our results or present different views. Moreover, the number of households was found to be better than the number of dwelling units as a proxy for population in this paper [1]. However, the number of households measures the designed households or the largest number of households, which are slightly different from real households in practice. Thus, some other proxies should be further explored and discussed to better represent population. In addition, extensive applications of our approach should be explored in the future. For example, as fundamental data of a city, urban population distribution data can be used to study social equity and environmental justice, conduct disaster assessments and site selection at a fine scale.

## 6. Conclusions

In this paper, we present a framework for estimating populations in residential neighborhoods with 30 m spatial resolution by integrating remote sensing and crowdsourcing data with little dependency upon census data. The number of households from a real estate trade platform was first employed as a proxy for population. The approach was tested in the urban core areas of Beijing. Some conclusions were drawn as follows. (1) We found that various and complex relationships between population and its related factors, including linear relationships, U-shaped relationships, and inverse U-shaped relationships. Compared with the relationships at a large scale, some of the linear relationships were similar, but there were weaker correlations at the residential neighborhood level. (2) Compared with NTL, location, and traffic, building features commonly had the highest importance for mapping urban population. (3) The number of households is a good proxy of population in a residential neighborhood. (4) Our framework exhibited a reasonable accuracy at residential neighborhood level, with a mean absolute percentage error of 47.26% and a $R^2$ of 0.78. The accuracy increased at the subdistrict level, with a mean absolute percentage error of 33.1% and a $R^2$ of 0.87. (5) Our framework could be used to renew the population data in real-time.

**Author Contributions:** W.Z. and C.J. designed the research. C.J. mapped population and analyzed the data. C.J., W.Z., Y.Q., and J.Y. wrote the paper together. All authors have read and agreed to the published version of the manuscript.

**Funding:** This research was funded by the National Natural Science Foundation of China, grant number 41771203; Chinese Academy of Sciences, grant number XDA23030102.

**Acknowledgments:** We acknowledge the contribution of Lijian Han, Jia Wang, and Jing Wang of Research Center for Eco-environmental Sciences for constructive comments.

**Conflicts of Interest:** The authors declare no conflict of interest.

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
