# Peer review of "Mapping the Urban Population in Residential Neighborhoods by Integrating Remote Sensing and Crowdsourcing Data"

_remotesensing, doi:10.3390/rs12193235_

Round 1

Reviewer 1 Report

I like the article. However, I see some critical points that should be addressed prior to publication.

1) I see a misleading title, or further clarifications on specific issues. The tile speaks about crowdsourcing data and Remote sensing. However, the article uses also the traditional population data from official statistics, if I understand well. So, please clarify pros & cons in the use of population data from different sources (counted in the census and estimated with RS or other sources). Please comment on the reliability of the different data sources, the intrinsic problems of data, the approximation of official statistics in respect of remote sensing, etc. You can not demise that all over the world, the basic source of information  for resident population is still official statistics from the census. So your contribution can go in the right direction if a broad discussion on these issues is opened and offered to readers.

2) I suggest enlarging the state of the art with other studies on a larger spatial scale, continents or global level. Please tell us the most recent advances in this field globally.

3) English usage is good, but I suggest to shorten some phrases in the abstract and discussion, because they seem to be rather long.

Author Response

1)I see a misleading title, or further clarifications on specific issues. The title speaks about crowdsourcing data and Remote sensing. However, the article uses also the traditional population data from official statistics, if I understand well.

So, please clarify pros & cons in the use of population data from different sources (counted in the census and estimated with RS or other sources). the reliability of the different data sources the intrinsic problems of data the approximation of official statistics in respect of remote sensing

You can not demise that all over the world, the basic source of information for resident population is still official statistics from the census. So your contribution can go in the right direction if a broad discussion on these issues is opened and offered to readers.

Responses: Sorry for the misleading title. Our title focused on a statistical modeling approach to mapping urban population. The statistical modeling approach is the core of our paper. We did use traditional population data from official statistics (census data). But the census data are mainly used in daymetric mapping approach (a comparative approach with the statistical modeling approach). So we didn’t show “census” in the title.

We agree with the reviewer that clarifying pros & cons in the use of population data from different sources is very important for improving our paper. So, we have added discussion about population data from different sources [Page 15-16, Line 558-585, with track changes].

2)I suggest enlarging the state of the art with other studies on a larger spatial scale, continents or global level. Please tell us the most recent advances in this field globally.

Responses: Thanks for your advice for enlarging the state of the art. Following the suggestion, we added some additional background about the advances at large scale. [Page 2, Line105-109, with track changes]

3)English usage is good, but I suggest to shorten some phrases in the abstract and discussion, because they seem to be rather long.

Responses: Following the suggestion, we revised the text, and rephased the long sentences to short ones [Page 1, Line 16-30, with track changes] and [Page 13-16, Line 457-585, with track changes]

Reviewer 2 Report

The paper aspires to characterize human activities on low scale (i.e. with high spatial resolution) using remote sensing of night time lights. The vast statistical analyses were also conducted to inspect various types of relationships between population and population-related factors.

However, I feel there are two main difficulties with the paper. One of the main shortcomings is that the authors relate population with almost each property available, even if for many relations studied there is no real reason for a significant correlation. It seems like that a possibility to relate one parameter with others has been preferably pursued without any deeper insight to the problem. I also disagree with authors statement that high human activity is ALWAYS correlated with a high NTL value. City centers, decorative lights, lights of industrial centers or railway stations, etc. are normally bright and only scarce dependent on intensity of human activity. We also know that light emissions to high elevation angles detected at the satellite level may also strongly depend on ambient environment in which the city lights are situated. A large portion of upwardly directed light can be due to reflection from city structs including trees and green elements which importance constantly increases in modern cities. It should be understood that the-content may show seasonal behavior resulting in varying lumen output (see PNAS 116, 7712-7717, 2019). Looking to low statistical indicators one can expect that seasonal variability of green content can change the trends for some of evaluated relationships. The above notes might be of even higher significance because VIIRS is not sensitive to the blue content of visible spectrum. Regarding some of the authors statements (e.g. on the line 355) I would like to emphasize that a new paper has been published recently in MNRAS Letters (496, L138-L141, 2020) showing that the correlation between population and NTL is not as high as expected, so the light pollution models should not rely much on census data.

The authors need to provide enough evidence for the importance of each statistical analysis performed, or should exclude those which have no theoretical foundation. Also a direct or indirect validation using other data source should be provided.

Author Response

1)One of the main shortcomings is that the authors relate population with almost each property available, even if for many relations studied there is no real reason for a significant correlation. It seems like that a possibility to relate one parameter with others has been preferably pursued without any deeper insight to the problem.

Responses: Thanks for your advice. These five aspects were selected based on their potential impacts on population distribution/density, as reported from previous studies, as well as data availability. Following the reviewer comment, we revised the text to explain and justify the selection of the factors. [Page 6, Line 256-264, with track changes]

2)I also disagree with authors statement that high human activity is ALWAYS correlated with a high NTL value. City centers, decorative lights, lights of industrial centers or railway stations, etc. are normally bright and only scarce dependent on intensity of human activity.

Responses: We agree with the reviewer, and changed “always” to “more likely”. [Page 3, Line 180, with track changes]

3) We also know that light emissions to high elevation angles detected at the satellite level may also strongly depend on ambient environment in which the city lights are situated. A large portion of upwardly directed light can be due to reflection from city structs including trees and green elements which importance constantly increases in modern cities. It should be understood that the-content may show seasonal behavior resulting in varying lumen output (see PNAS 116, 7712-7717, 2019). Looking to low statistical indicators one can expect that seasonal variability of green content can change the trends for some of evaluated relationships. The above notes might be of even higher significance because VIIRS is not sensitive to the blue content of visible spectrum.

Responses: We totally agree with the reviewer, and thereby change the text for clarifications. We added a sentence and a reference [Page 13, Line462-463, with track changes].

Reference: Kocifaj, M.; Solano-Lamphar, H.A.; Videen, G. Night-sky radiometry can revolutionize the characterization of light-pollution sources globally. Proceedings of the National Academy of Sciences 2019, 116, 7712,doi:10.1073/pnas.1900153116.

4)Regarding some of the authors statements (e.g. on the line 355) I would like to emphasize that a new paper has been published recently in MNRAS Letters (496, L138-L141, 2020) showing that the correlation between population and NTL is not as high as expected, so the light pollution models should not rely much on census data.

Responses: We have read the recently published paper, and revised the sentence by replacing “there is no debate that” with” generally”(Page 13, Line 457-460, with track changes) to reflect the findings from this paper. We also citied this paper to support the statement.

Reference: Kocifaj, M.; Komar, L.; Lamphar, H.; Wallner, S. Are population-based models advantageous in estimating the lumen outputs from light-pollution sources? Monthly Notices of the Royal Astronomical Society 2020, 496, L138-L141, doi:10.1093/mnrasl/slaa100.

Round 2

Reviewer 2 Report

The authors improved the paper upon review comments. I have no other critical comments, however I am still a bit skeptical regarding vague correlation and statistical analysis. Nevertheless, I am inclined to accept the paper to make the work publicly available to a broad scientific community for further commentary.